# Worldwide Overview and Country Differences in Metaverse Research: A Bibliometric Analysis

**Jinlu Shen [1], Xiangyu Zhou [1], Wei Wu [1,2,*], Liang Wang [1] and Zhenying Chen [3]**

[1]  School of Public Affairs, Zhejiang University, Hangzhou 310058, China
[2]  Institute of China's Science, Technology and Education Policy (iCSTEP), Zhejiang University, Hangzhou 310058, China
[3]  Library, Zhejiang University, Hangzhou 310058, China
*   Correspondence: ww2015@zju.edu.cn; Tel.: +86-135-6712-2290

**Abstract:** As a research topic integrating various subjects and technologies, Metaverse research has been a global concern in recent years. This study explores the hotspots and frontiers of academic research on the Metaverse based on a bibliometric analysis from 2012 to 2021. A keyword retrieval dataset related to Metaverse research was constructed based on expert consultation and manual reading of the literature, retrieving articles and conference papers from the Scopus database. Critical points in Metaverse research are represented in terms of research scale, keyword co-occurrence networks, keyword citation bursts, and international collaborative networks with the application of VOSviewer and CiteSpace's bibliometric visualization software. The results indicate that Metaverse research is experiencing rapid growth, with countries/regions increasing their production at varying speeds. The results also indicate the three most prolific countries, the United States, China and Germany, for comparison, showing the leading topic as virtual reality in Metaverse research, and we find that there are differences in topic clustering and hotspot evolution among the three countries over the past decade. By determining the current research status and the overall development path of the Metaverse field, the paper intends to provide a reference for the future research development and technical application of the Metaverse.

**Keywords:** Metaverse; bibliometric analysis; VOSviewer; CiteSpace

## 1. Introduction

The term "Metaverse" was first introduced in the science fiction novel "Snow Crash", published by American science fiction writer Neal Stevenson in 1992, and was considered to be connected with the real world by envisioning a virtual-reality-based successor to the Internet [1]. The prefix "Meta" means "transcendence" and "verse" refers to the world and universe [2]. In terms of a historical view, before the concept of the Metaverse was proposed, human beings were engaged in a large number of productive labor and social activities through the Internet human–machine interface, thus creating an invisible but effective social structure. With the rapid development of a series of technologies, such as virtual reality, augmented reality and sensors, the concept of a digital social structure has been continuously developed and refined, and it has been gradually transformed into today's "Metaverse", which is considered as the infrastructure of the next-generation Internet [3,4].

In the development of the Metaverse, the game field is one of the most important driving forces. In 2003, the online virtual game "Second Life" was introduced, which was, to a large extent, the predecessor of the Metaverse, and a combination of online games, social platforms and Web 2.0. The popularity of Second Life has not only been due to the inexpensive participation, but mainly due to the opportunity of providing the "residents" (players) with the opportunity to engage in any desired activity, such

as gathering, wandering, driving, eating, traveling, manufacturing and even shopping outside of the game [5]. Due to the game's early development, Second Life was later criticized for failing to provide a good user interface in which players could easily move and interact within the user-built worlds [6]. In 2006, Roblox was introduced. There was no plot in Roblox; it was completely built by the players themselves and they could develop their own ideas into new game modules within the game and sell them through virtual currency. Roblox has also been used as an educational tool in the classroom for motivation, problem-solving and STEM [7]. In 2016, Pokémon GO was released as the first game that superimposed the virtual world onto the real world, and it quickly became popular around the world. Growing up during a period of rapid evolution in terms of PC and mobile technologies and games, one of Generation Z's defining characteristics is the preference for game-like experiences, and this has made them the main users of the Metaverse [2].

Another case that cannot be ignored in the field of the Metaverse is the promotion of the blockchain concept and the virtual currency built upon it. As a combination of cryptography, game theory and peer-to-peer networking without central co-ordination [8], blockchain is one of the domain technologies that has led to a recent growth in interest in the Metaverse. In 2009, Satoshi Nakamoto, the founder of Bitcoin, excavated the origin block of Bitcoin (block number 0) and produced 50 Bitcoins. In the same year, Nakamoto tried to use blockchain as the public transaction ledger of Bitcoin. In 2012, in a project led by Vitalik Buterin, the concept of Non-Fungible Tokens (NFT) came into being, defined as digital pure assets that cannot be exchanged like-for-like (equivalently, non-fungible) [9]. This is the only cryptocurrency token used to represent digital assets, as the first application of blockchain technology to achieve clear public prominence in early 2021 [10]. It is irreplaceable but can be bought and sold. In 2015, Vitalik Buterin and Gavin Wood launched the Ethereum network and Ethereum blockchain, and then built Decentraland, the first fully decentralized and user-owned virtual world based on the Ethereum blockchain. The price series of these NFTs have been characterized by inefficiency and a rise in value [11].

In recent years, blockchain and virtual currency have become more familiar and begun to be further integrated with the game field, with games having incorporated their own economies, commerce and currencies [12]. Data analysis related to 6.1 million trades of 4.7 million NFTs, primarily from the Ethereum and WAX blockchains, indicated that the most exchanged NFTs belonged to the gaming industry [13]. In 2018, the Dai stable currency was released on Ethereum, which added a new element to the turbulent cryptocurrency world. Compared with other cryptocurrencies, the centralized Dai stable currency is linked to the US dollar, making it much less volatile and more reliable for Decentralized Finance (DeFi). In the same year, the blockchain game Axie Infinity, a type of digital universe, was launched. Many players from Vietnam and the Philippines quit their jobs and devoted themselves to playing the game every day, hoping to gain more wealth. The existence of limited volatility transmission effects was identified between NFT pricing (Decentraland, Cryptopunks and Axie Infinity) and cryptocurrencies (Bitcoin and Ethereum) [10]. It has been predicted that gamers, traders and companies will be attracted by the growth of a new niche in the cryptocurrency market, as well as the absence of dependence on the cryptocurrency market [12].

The COVID-19 pandemic, beginning in 2020, profoundly changed the global pattern and social operation mode, leading to a "new normal" [14]. The core feature of the Metaverse has gradually evolved to take entertainment applications as a priority and commercial operation as the main battlefield, and various institutions have built more abundant application scenarios by applying the concept of the Metaverse with obvious characteristics of commercial operation. Various types of startups and capital groups are constantly entering the market and seeking new rounds of transformation opportunities, and they successively promote the implementation of related applications. In July 2021, Mark Zuckerberg, the CEO of Facebook, announced that Facebook would be renamed Meta (initially named Facebook Horizon Worlds), changing from a social media company to a Metaverse company [6]. In fact, Facebook has long been looking forward to the layout of

Metaverse technology-related patents, and it acquired Oculus, a virtual reality headgear manufacturer, in 2014. Another new opportunity for businesses is to communicate with clients directly through avatars instead of depending on traditional marketing tactics [15]. For instance, Jensen Huang, the founder and CEO of NVIDIA, used a virtual avatar to launch new products, instead of doing so himself, at the Global Traffic Conference (GTC) in April 2022. A concern among researchers is that, as the involvement of large tech companies such as Meta, Google, Apple and Microsoft during the COVID-19 pandemic has created the effect and motivation of legitimate surveillance capitalism, data privacy risks have increased and may lead to privacy encroachment and mind control [14].

The COVID-19 pandemic has accelerated the integration of the healthcare field with cutting-edge technologies, and the desire to break the limitations of time, distance and technology as much as possible, which has led to the widespread application of Metaverse applications in the healthcare field. Metaverse technology also supports infrastructure and opportunities for enhancing the education of health care workers. The mechanical resolution of the sense of touch awarded the Nobel Peace Prize in Physics in 2021 was regarded as one of the most significant milestones for the field [16]. Concepts such as *Spine Metaverse* began to emerge, serving as the care delivery between a real-life patient in an actual hospital and a remote surgeon operating from their own home base, as well as a natural focal point for organically grown visual displays of large-scale data collections [17]. The bibliometric analysis has been used in certain literature, resulting in the Metaverse being able to be adopted for diagnostic and surgical procedures and rehabilitation for pain, stroke, anxiety, depression, fear, cancer, and neurodegenerative disorders with satisfying results [18]. Furthermore, VR-aided therapy was identified into four major research areas— post-traumatic stress disorder (PTSD), anxiety- and fear-related disorder (A&F), diseases of the nervous system (DNS), and pain management [19].

In the meantime, in the field of education, teaching and learning activities began to consider their combination with the Metaverse scene. For example, UC Berkeley set up a "virtual campus" in the game "My World", allowing students to participate in the graduation ceremony in the form of "virtual avatars". In June 2020, at the Chinese University of Hong Kong, Shenzhen (CUHKSZ) Metaverse, the implemented blockchain-driven university campus prototype, was illustrated to effectively enrich the campus life of university students and university faculties, with students' actions in the real world correspondingly affecting the virtual world [3]. The Computer Department of Tsinghua University held the "Huazhibing" achievement conference in June 2021. As the first original virtual student in China, Huazhibing started her study and research career at Tsinghua University. Students can interact with teachers and communicate with classmates through their avatars, indicating one of the advantages of the Metaverse, which is to create an immersive learning opportunity, helping to enhance students' learning motivation [1]. It was also inferred that using the Metaverse with augmented reality in the teaching of mathematics significantly favored students' learning performance in high school [20]. Furthermore, a bottom-up approach was employed to identify five world types, i.e., survival world, maze world, multi-choice world, racing/jump world and escape room world, that can be used for metaverse-based education [2].

In terms of the design of virtual inhabitable cities or digital twin cities using large-scale data-driven AI systems, regarding the Metaverse, there has also been a key trend in smart urbanism [14,21]. In November 2021, Seoul, the capital of South Korea, stated that it would become the first major city in the world to enter the Metaverse. The fully operational Metaverse platform would provide various public services and cultural activities in 2026, and the city has already launched a smartphone pilot. Dubai launched a number of Metaverse ventures in 2022. Urban public services have been applied with the Metaverse—for example, using blockchain data platform Chainalysis to offer virtual training programs to government agencies. The Dubai government proposed the "Dubai Metaverse Strategy" in July 2022, which aims at creating 40,000 jobs and adding 4 billion dollars by 2030 to the country's economy over the next five years. As a technological vision, the peculiar

characteristics of the experience of everyday life in the data-driven smart cities of the future have been depicted by the rise of the Metaverse [4].

The application of the Metaverse has promoted the rise of related research topics. The architecture of the Metaverse, however, has not reached a consistent definition. A seven-layer Metaverse, including architecture infrastructure, a human interface, decentralization, spatial computing, a creator economy, discovery and experience, was proposed by Jon Radoff, while another three-layer metaverse of architecture, infrastructure, interaction and ecosystem from bottom to top was considered from a more macro perspective [3]. Combining the two critical uncertainties, the spectrum of technologies and applications, ranging from augmentation to simulation, and the spectrum ranging from intimate (identity-focused) to external (world-focused), a Metaverse road map was introduced that categorized key components of the Metaverse future into four types: virtual worlds, mirror worlds, augmented reality and lifelogging [22]. These provide a user experience based on a 3D graphic environment and a studio program, serving as a world development tool [2]. A conceptual model with five interacting components was constructed to be fundamental for understanding teamwork in a Metaverse environment, which were the metaverse itself, people/avatars, metaverse technology capabilities, behaviors and outcomes [23].

An earlier definition of the Metaverse was given as an "immersive three-dimensional virtual worlds where people interact with each other and their environment, using the metaphor of the real world but without its physical limitations" [24]. With the rapid growth and influence of the Metaverse, especially amid the COVID-19 pandemic, a new definition is needed compared to the early versions of the Metaverse [1]. This paper defines the Metaverse as a new Internet application and social form that integrates physical space technology, virtual space technology, virtual–real interaction technology and other new technologies. It builds a physical space foundation based on 5G/6G Networks, Internet of Things, cloud computing and edge computing, etc. [14]. The immersive virtual environment is constructed based on digital twins, modeling and simulation technologies. The physical space and virtual world are connected based on mixed reality, voice interaction, tactile feedback, brain–computer interfaces and other technologies. Based on blockchain, security, trust, privacy computing and other technologies, it provides identity authentication, guarantees data security and user privacy and maintains a credible virtual and real world. To conclude, the Metaverse is not a single technology, or merely a game-playing method, but a combination and upgrading of the socio-technical imaginaries [25]. The rise of the Metaverse is the result of the superposition of three waves.

The first is the globalization of human sensory stimulation. Throughout the historical process of humans receiving and processing information, it is an irreversible evolutionary direction from single sensory stimuli such as books, paintings, sculptures and music to multiple sensory stimuli, and then to the global multi-sensory stimuli of the Metaverse. Secondly, it involves the integration of cutting-edge technology applications. The Metaverse is a new type of Internet application and social form integrating a variety of new technologies, and the integration of many cutting-edge technologies provides technological possibilities for the incubation of the Metaverse [26,27]. It provides an immersive experience based on extended reality and brain–computer interface technology, generating a real-world image based on digital twin technology, building an economic system based on blockchain technology and creating new content in real time based on AI technology. It closely links the virtual world to the real world in an economic system, social system and identity system, and it provides each user with equal opportunities for participation [7]. The third is the virtualization of residents' daily lives. Since the peak of the COVID-19 pandemic, the scale of social networking has increased significantly, and the "stay-at-home economy" has developed rapidly. Online life has changed from the original short-term exception state to a normal state, as well as from being a supplement to the real world to a digital twin world, while the Metaverse is an upgraded version of the existing cyberspace.

Under the combination of the above three waves, the Metaverse has become the best intersection of new products driven by technology as well as new experiences expected by users, reflecting an inevitable trend of digital and technological progress. With the rise and vigorous development of the Metaverse topic, bibliometric research focusing on a certain perspective, such as the "Health Metaverse" and "Metaverse in education", has been conducted in the past year or two [1,4]. However, an overall visual bibliometric analysis is still rare. This paper presents a bibliometric analysis of the Metaverse, including an analysis of the identification and study of the most productive scientific structures (major authors, sources, institutes and countries) and international collaborations from 2012 to 2021, as well as a comparison of institutions' performance, hotspots and bursts over the past decade, using VOSviewer and CiteSpace software, respectively, to perform visualization.

## 2. Materials and Tools

### 2.1. Data Collection

The methodology applied in this study was bibliometric analysis. Bibliometric analysis is a quantitative application to study the characteristics, growth and maturation of scientific production [28,29], and it includes geographical or institutional aspects and indicators of developing performance over specific time periods [30]. Bibliometric studies are available in Metaverse-related technical fields such as virtual reality [31,32], blockchain [8,33] and cloud computing [34,35]. In fact, as a research field involving a wide range of disciplines and technologies, many Metaverse-based technologies have long been known to the public. As academic research results have been closely related to the development of the Metaverse for years, they have become an indispensable form of content in the study and analysis of the Metaverse area. Therefore, in the process of obtaining the Metaverse research literature, we should not only consider the core conceptual vocabulary, but also include the technical fields closely related to its formation and development. Whether the keyword selection is complete and whether the retrieval construction is reasonable has a crucial impact on the accuracy and credibility of the Metaverse bibliometric analysis results.

In view of the wide range of disciplines and technical fields involved in the Metaverse area, a large number of keywords were initially screened by the research group through manually reading the important literature. The scope of literature reading included key reports such as White Paper on China's Metaverse edited by Gong, review literatures [19,36–38] and other key literatures published by representative institutions [39–41], etc. In the meantime, this study combined keyword clustering to extract high-frequency words and used the database trend analysis function to supplement keywords. In order to ensure the accuracy and comprehensiveness of the keyword set, experts in different technical fields related to the keywords were fully consulted, and the keyword set was constantly adjusted and optimized through trial tests based on expert opinions.

On the basis of multiple iterations of the above two aspects, the search expression was finally determined to ensure that the search results could objectively reflect the research status of related fields. "Metaverse", "Meta universe", "Metauniverse", "Meta-cosmic", "Metacosmum" and "virtual universe" were the core keywords chosen. Related research technology areas were included to allow for the expansion of the Metaverse focus with the terms, which were Extended Reality, Virtual Reality, Augmented Reality, Mixed Reality, Digital Twins, Blockchain/Blockchains, 5G Technology, 5G Networks/6G Technology/6G Networks, Intelligent Networks, Sensor Networks, Internet of Things, Cloud Computing, Edge Computing, Intelligent Computing, Virtual Human, Digital Human, 3D Modeling, Computer Animation, Game Engine, Computer Vision, Computer Graphics, User Generated Content, Helmet Mounted Display/Head Mounted Display, Hand-Held Input/Control Device, Non Hand-Held Input/Control Device, Motion Input/Control Device, Brain Computer Interface and Non-Fungible Token. The specific search process also made use of the subordinate keywords of the broad technical category—for example, the subordinate words of Motion Input/Control Device include "On-body User Interaction",

"Haptic Feedback", "Simulated Haptic Cues", "Human Pose Tracking", "Eye Tracking", "Action Recognition" and "Motion Capture". In general, the most important supporting technologies of the Metaverse were divided into physical space technology, virtual space technology, virtual–real interaction technology and virtual–real generic technology [36]; they are presented in Table 1, which shows the selection of specific keywords.

**Table 1.** Keywords related to Metaverse research.

| Category | Keywords |
|---|---|
| Metaverse itself | metaverse, meta universe, metauniverse, meta-cosmic, metacosmum |
| Metaverse perspectives | extended reality, virtual reality, augmented reality, mixed reality, reality virtuality, virtual-real fusion, virtual world, immersive internet, virtual space, virtual environment, virtual service provider, immersive experience, immersive feeling, immersive environment, virtual environment |
| Physical space technology | cyber world, physical world, 5G mobile, 5G technology, 5G network, 6G technology, 6G network, network awareness, perception network, user centric network, user centric networks, human centric network, human centric networks, software defined network, software defined networking, software defined networks, self-organizing network, cloud computing, edge computing, edge intelligence, internet of things, robots, collaborative robot, human–robot collaboration, autonomous driving, autonomous cars, autonomous vehicles, connected vehicles, mobile broadband, enhanced mobile broadband, semantic communication network, semantic communication, intelligent network, brain-inspired intelligent, brain-inspired intelligence, computing power network, graphics processing unit, sensor network, cyber physical system, data storage, data sharing, data interoperability, edge cloud collaboration, cloud edge cooperation, hybrid edge cloud, integrated sensing communication |
| Virtual space technology | digital twin, avatar, digital human, virtual human, virtual character, virtual agent, 3D modeling, three-dimension modeling, virtual simulation, image segmentation, image restoration, image enhancement, image inpainting, image generation, real-time rendering, 3D rendering, computer animation, digital economy, digital finance, virtual economy, Non-Fungible Token, decentralized finance, cryptocurrency, digital currency, ethereum, smart contract, distributed ledger, virtual property, virtual asset, digital transaction, digital identity, digital asset, digital right, game engine, user generated content |
| Virtual–real interaction technology | head mounted display, hand-based input device, non-hand-based input device, helmet mounted display, motion input device, mobile headset, ARheadset, VR headset, google glass, Microsoft hololens, wearable device, user interface, graphical user interface, brain computer interface, mid-air pointing, floating icon, floating menu, freehand interaction, interaction paradigm, human computer interaction, electroencephalography, electromyography, IMU-driven user interaction, on-body user interaction, haptic feedback, object weighting, virtual spring, simulated haptic cues, just noticeable difference, tactile internet, telepresence, holographic display, holograph, 3D aerial hologram, glasses-free 3D, autostereoscopy, human pose tracking, eye tracking, scene understanding, semantic segmentation, object detection, stereo depth estimation, action recognition, gesture recognition, palmprint recognition, environmental perception, human–IoT interaction, virtual landscape, oculus quest, motion capture, brain machine interface, natural language processing, computer vision, simultaneous localisation mapping, voice input/output |
| Virtual–real generic technology | privacy social acceptability, blockchain, distributed system, decentralized autonomous organization, distributed database, decentralized identifier, verifiable credential, finite state machine, reinforcement learning, machine learning, artificial intelligence, deep learning, resource management, resource allocation, big data, perception, security, trust, authentication, coordination |

It was proven that both the number of papers and the number of citations received by countries, as well as broken down by field, were extremely high in the Web of Science database and the Scopus database [42,43], while Scopus showed larger journal coverage in all fields [44]. Therefore, the Elsevier Scopus database was chosen in our research, and the timespan covered the years 2012–2021 to identify the key points in developing Metaverse–related research over the past decade. The reason for choosing to search since 2012 is that the first Non-Fungible Token (NFT) Colored Coins in history came out, serving as an economic bridge between Metaverse and the real world, leading the increasing integration of the real world and the digital world [7]. The emergence of NFTs enables the Metaverse to expand various social meanings (such as fashion, activities, games, educa-

tion, office, etc.) on the basis of immersive interaction, which is different from the early concept of the Metaverse [45]. Descriptors were subject to a triangular search (title–abstract–keywords), considering the identification of the central focus of the Metaverse research [29]. The search was confined to the literature published and extracted by 31 December 2021, and only the Article and Conference Paper types were included in this review. Thus, a total of 45,178 scientific documents were retrieved from the Scopus database as the data basis for the bibliometric analysis, containing abundant information such as the publication year, authors and their affiliations, title, abstract, source journal, subject categories and references, etc. For convenience, we will refer to Metaverse-related fields of research as "Metaverse" research in the following sections.

### 2.2. Methodology Tool

Performance analysis and science mapping analysis are two major fields of study concerning bibliometrics. Performance analysis aims to evaluate different scientific actors, such as researchers, institutions and countries, while science mapping analysis relies on the topological and temporal representation of the cognitive and social structure of a particular research field [28]. As an important research topic, bibliometric mapping can also be distinguished as two aspects, one for the construction of bibliometric maps and the other for the graphical representation of such maps [46]. In this study, a knowledge map was used for bibliometric analysis, and two different methodology tools were chosen.

The first tool was VOSviewer. VOSviewer is a program used for constructing and viewing bibliometric maps [46], which can present network visualization, overlay visualization, density visualization and handle a large number of items. It uses a "clustering technique" to cluster publications and further analyze the resulting clustering solutions [47]. Network visualization was chosen in the study to illustrate the keyword co-occurrence network in the Metaverse field from 2012 to 2021, both globally and in the United States, China and Germany for comparison. Co-author-based transnational collaborative network visualization was also generated by VOSviewer.

CiteSpace, another methodology tool used in the study, was jointly developed by Chen Chaomei from the School of Information Science and Technology of Drexel University and the WISE Laboratory of Dalian University of Technology [48]. It is a Java application supporting visual exploration with knowledge discovery in bibliographic databases [49], playing an instrumental role in detecting emerging trends and abrupt changes in a timely manner, which cannot otherwise be easily captured using other tools [50]. Therefore, this paper presents the evolution trend of global research hotspots by using the analysis of keyword citation bursts in CiteSpace, and it compares the research frontiers in the field of the Metaverse among the United States, China and Germany. The main analytical procedure for the bibliometric analysis of Metaverse research is displayed in Figure 1.

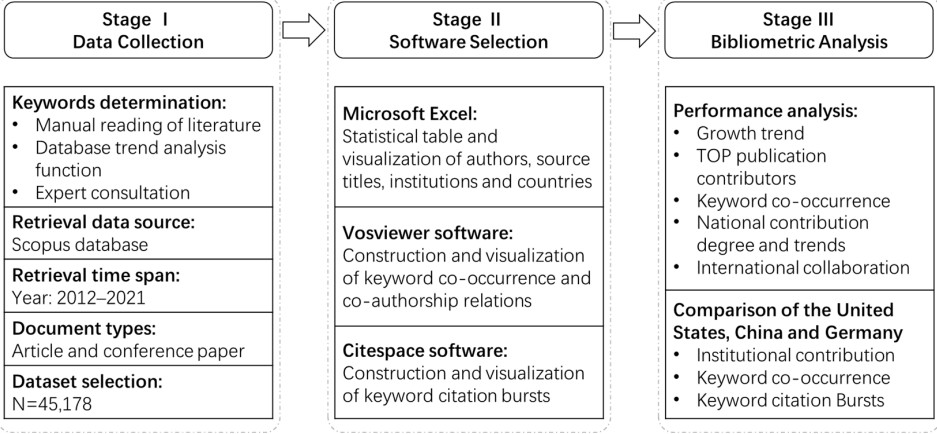

**Figure 1.** Process of bibliometric analysis relating to Metaverse.

## 3. Results

### 3.1. General Publication Output and Growth Trend

According to the retrieval results of Scopus, a total of 45,178 scientific documents met the exclusion and inclusion criteria, distributed between 2012 and 2021, more than 60% of which were conference papers. Figure 2 presents the number of documents per year, and the proportion of conference papers and articles. In each decade, conference papers outnumbered articles, indicating that conference papers are important carriers of the research achievements related to the Metaverse. Scientific production in the Metaverse has shown a growing trend over the past decade, from 2528 documents in 2012 to over 7000 documents in 2021. It should be noted that the number of documents in 2020 and 2021 was lower than that in 2019, which may be due to the cancellation or postponement of some conferences or the lag in database inclusion, and the data of the two years still show a continuous increasing trend at present.

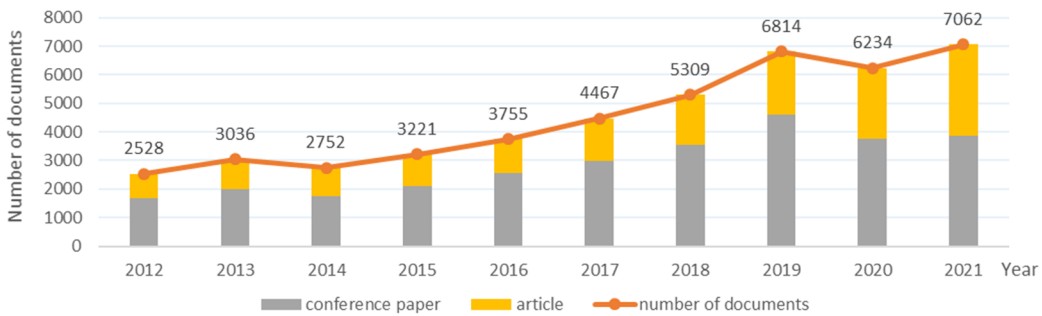

**Figure 2.** Growth of scientific publications relating to the Metaverse.

The growing performance for the publication output may reflect both the motivation of researchers' own interests and the motivation of social demands, and particularly the rapid development of artificial intelligence, cloud computing and other technologies. To further demonstrate the greatest contributors to the publications in the Metaverse field over the 10-year period, the 10 most productive authors, sources and institutions were obtained from the Scopus database, as shown in Table 2; the numbers in parentheses all indicate the number of publications over a 10-year period.

**Table 2.** Top 10 publication contributors related to the Metaverse.

| Ranking | Authors | Source Title | Institution |
|---|---|---|---|
| 1 | Billinghurst, M. (107) | Lecture Notes In Computer Science Including Subseries Lecture Notes In Artificial Intelligence And Lecture Notes In Bioinformatics (3060) | Chinese Academy of Sciences (460) |
| 2 | Steinicke, F. (93) | ACM International Conference Proceeding Series (1270) | Centre National de la Recherche Scientifique (446) |
| 3 | Latoschik, M.E. (78) | Advances In Intelligent Systems And Computing (796) | Beihang University (330) |
| 4 | Bruder, G. (77) | Conference On Human Factors In Computing Systems Proceedings (649) | Technical University of Munich (330) |
| 5 | Narumi, T. (70) | Communications In Computer And Information Science (601) | The University of Tokyo (324) |
| 6 | Slater, M. (69) | IEEE Access (457) | University of Southern California (295) |
| 7 | Hirose, M. (65) | Proceedings Of SPIE The International Society For Optical Engineering (392) | Beijing University of Posts and Telecommunications (278) |
| 8 | Navab, N. (59) | Journal Of Physics Conference Series (318) | Shanghai Jiao Tong University (275) |
| 9 | Woo, W. (53) | Proceedings Of The ACM Symposium On Virtual Reality Software And Technology VRST (317) | University of Central Florida (273) |
| 10 | Kiyokawa, K. (51) | 26th IEEE Conference On Virtual Reality And 3D User Interfaces VR 2019 Proceedings (300) | Tsinghua University (265) |

Among the contributors, Billinghurst M, an author from the University of South Australia, has published the most work in the Metaverse field over the last decade; users' body gestures and interaction repertoire for wearable displays in the field of augmented reality (AR) are the most cited papers [51–53], and are all in collaboration with Piumsomboon T. The second most highly contributing author is Steinicke F from Universität Hamburg, who has linked the research of immersive virtual reality with head-mounted displays (HMDs) [54–56]. Latoschik M.E. from Julius-Maximilians-Universität Würzburg is also interested in head-mounted displays, contributing to the value of personalized avatars as well as the effect of avatar realism [57,58].

Lecture Notes In Computer Science Including Subseries Lecture Notes In Artificial Intelligence And Lecture Notes In Bioinformatics, as the top publisher in the Metaverse area, had over three thousand publications from 2012 to 2021, more than twice as many as the runner-up. The most cited paper from this journal proposed a novel deep learning framework for multivariate time-series classification in 2014 [59]. As listed in Table 2, the conference paper is an important source of publication for Metaverse research; for example, the ACM International Conference and the Conference On Human Factors In Computing Systems rank second and fourth among sources and have contributed 1270 and 796 articles, respectively.

In terms of the institutional performance in the Metaverse area, the first and second positions are occupied by academic institutions, which are the Chinese Academy of Sciences (CAS) from China and the Centre National de la Recherche Scientifique (CNRS) from France. With the exception of these two highly contributing institutions, the other eight are all universities. China has the largest share of the eight universities, which are Beihang University, Beijing University of Posts and Telecommunications and Tsinghua University. The University of Southern California, as well as the University of Central Florida, are in the United States. The remaining two are the Technical University of Munich in Germany and the University of Tokyo in Japan.

### 3.2. Keyword Analysis

Keyword analysis has been used for document categorization, summarization, indexing and clustering in many domains, and the keyword-based search is becoming an essential means to identify relevant information. We used keyword co-occurrence analysis via VOSviewer in this study [60]. In total, 132,017 keywords were included, and the co-occurrence threshold of the keywords was set as 200. Due to a certain number of acronyms in the field of Metaverse research, some of the synonyms were replaced and merged in VOSviewer, such as human–computer interaction and HCI, Internet of Things and IOT, and software-defined networking and SDN. In total, 401 items were finally brought into visualization, as shown in Figure 3, and the top 10 occurrences of keywords are listed in Table 3.

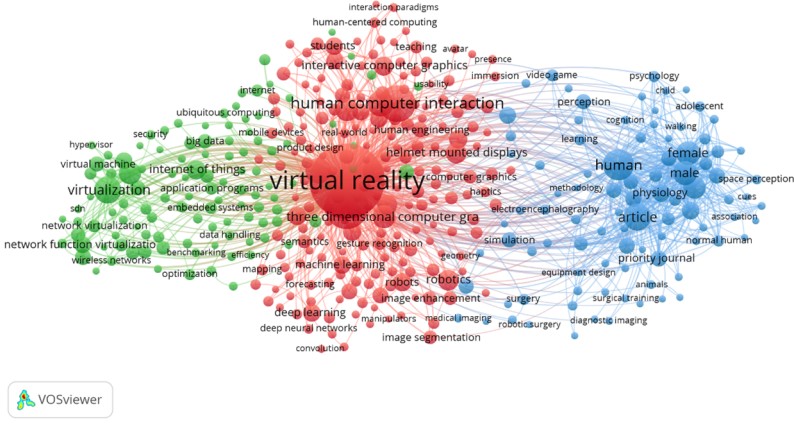

**Figure 3.** Co-keyword network visualization related to the Metaverse.

**Table 3.** Top 10 occurrences of keywords related to the Metaverse.

| Keyword | Cluster | Links | Total Link Strength | Occurrences |
|---|---|---|---|---|
| virtual reality | 1 | 400 | 164,198 | 30,794 |
| augmented reality | 1 | 399 | 48,526 | 9829 |
| human–computer interaction | 1 | 396 | 33,758 | 6534 |
| human | 3 | 384 | 51,476 | 4652 |
| article | 3 | 380 | 41,050 | 3532 |
| virtualization | 2 | 282 | 19,322 | 3482 |
| user interfaces | 1 | 387 | 16,430 | 3133 |
| artificial intelligence | 1 | 400 | 15,200 | 3054 |
| male | 3 | 336 | 34,703 | 2643 |
| female | 3 | 337 | 33,534 | 2551 |

As shown in Figure 3, a keyword co-occurrence network was generated, using the size of each circle to represent the frequency of keyword occurrences. The distances among the circles, connected with curves, represent the relevance of the keywords. The largest circle in the figure is labelled by the keyword "virtual reality", which also represents the highest number of occurrences, as well as the highest total link strength, as demonstrated in Table 3. According to the VOSviewer manual, each link has a strength represented by a positive numerical value [61]. "Total link strength" refers to the total number of co-occurrences of one keyword and other keywords, including the number of repeated co-occurrences. As Table 3 shows, the total link strength of the keyword "virtual reality" is far superior to that of all other keywords, reaching 164,198.

Figure 3 is clearly color-divided into three clusters. Circles in the same color cluster share a similar topic among publications in the Metaverse field, with three distinct clusters illustrated. Among these clusters, the red one links the most keywords, with the number of 197, led by a "center" keyword, which is virtual reality. The words with high occurrence are augmented reality, human computer interaction, user interfaces, three-dimensional computer graphics, artificial intelligence, e-learning, robotics, helmet mounted displays and visualization, which generally refer to the topic "human-computer interaction". In the cluster colored green, the main keywords in the network virtualization are cloud computing, internet of things, network function virtualization, network security, transfer functions, decision making, distributed computer systems, virtual machine, software defined networking, etc., which are more concerned with "computer and network". Another central cluster, highlighted in blue, comprised keywords such as human, male, female, movement, clinical competence, surgical training, robotics surgery, patient rehabilitation and computer assisted surgery, which mainly concerned "new technologies in clinical medicine".

In this section, the analysis of keyword citation bursts is also introduced to explore the dynamics of Metaverse research, as well as the most intensively researched directions. Keyword citation bursts, which refer to keywords increasing sharply in citations, can reflect the changes and duration of a keyword in different years, and thus help researchers to determine the most prevalent content and development trends of this field. CiteSpace was used for keyword burst detection, and thus the emergence, development and decline in Metaverse research was illustrated in the period from 2012 to 2021, providing insights into the most dynamic and relevant research topics.

According to the analysis results of CiteSpace, 35 keywords have citation bursts, and the 20 most commonly detected hotspot keywords were retained, as shown in Figure 4. The light blue line segment indicates that the keyword has not appeared, and the dark blue denotes the time period of keyword appearance, while the red line segment indicates the time period of the keyword citation burst. Figure 4 also presents the strength of the keyword citation bursts. The greater the strength, the higher the possibility of becoming a research hotspot at that time. The highest degree of strength is presented by the keyword "virtualization", with the only strength value over 200, thus beginning to burst in 2012 and ending in 2017. It also shares the longest burst period with the keyword "network virtualization", whose strength is 100.49, further demonstrating that "virtualization" was an important research hotspot in the early years.

| Keywords | Year | Strength | Begin | End | 2012 – 2021 |
|---|---|---|---|---|---|
| virtualization | 2012 | 203.7 | **2012** | 2017 | |
| cloud computing | 2012 | 163.98 | **2012** | 2016 | |
| virtual world | 2012 | 143.98 | **2012** | 2016 | |
| algorithm | 2012 | 102.32 | **2012** | 2016 | |
| network virtualization | 2012 | 100.49 | **2012** | 2017 | |
| avatar | 2012 | 97.6 | **2012** | 2016 | |
| three dimensional | 2012 | 70.48 | **2012** | 2013 | |
| virtual environment | 2012 | 69.88 | **2012** | 2015 | |
| distributed computer system | 2015 | 73.54 | **2015** | 2017 | |
| education | 2015 | 66.62 | **2015** | 2017 | |
| network function virtualization | 2016 | 132.15 | **2016** | 2019 | |
| software defined networking | 2016 | 92.64 | **2016** | 2018 | |
| transfer function | 2017 | 66.28 | **2017** | 2018 | |
| network security | 2017 | 65.79 | **2017** | 2018 | |
| edge computing | 2018 | 73.68 | **2018** | 2019 | |
| deep learning | 2018 | 125.42 | **2019** | 2021 | |
| human computer interaction (hci) | 2019 | 81.97 | **2019** | 2021 | |
| user experience | 2019 | 80.26 | **2019** | 2021 | |
| 3d modeling | 2013 | 78.4 | **2019** | 2021 | |
| vr | 2019 | 67.72 | **2019** | 2021 | |

**Figure 4.** Top 20 keywords with the strongest citation bursts related to the Metaverse.

Among these keywords, "cloud computing", "virtual world", "algorithm", "virtual environment", "avatar" and "three dimensional" were also earlier burst keywords, beginning in 2012. Specifically, "three dimensional" had the shortest burst time period, from 2012 to 2013. In more recent years, the keywords "human computer interaction(hci)", "user experience", "3D modeling", "vr" and "deep learning" began to burst in 2019 as the newest hotspot keywords, representing the continuation of major topics in recent years. This is consistent with the trend of strengthening the human–computer interaction and emphasizing the user experience under the study of the Metaverse.

### 3.3. Performance Analysis by Country and Region

3.3.1. Contributions by Country and Region

Recently, the Metaverse has become a new and relevant topic around the world. According to the search results, nearly 140 countries and regions in the world have participated in basic research related to the Metaverse, among which nearly 50 countries have more than 100 publications. There are 13 countries with more than 1000 documents, accounting for more than 80% of the world's publications. The United States and China are the two countries that have provided the most contributions in the area of the Metaverse, with a total number of 9201 and 7454, respectively. Germany ranks third in the number of publications—there is a relatively large gap between Germany and the United States and China regarding the number of publications. Other major contributors include the United Kingdom, Japan, Italy, South Korea and so on.

This study designs a national contribution index to measure the competitiveness of a country, which can better describe the contribution of relevant countries to Metaverse research from the context of global-research-scale changes. The national contribution refers to the proportion of the number of publications from a certain country to the number of global total publications. The statistical principle is that a country contributing to a document will be counted once, while a multinational collaborative document will be counted once for each country. Figure 5 shows the 10 countries with the greatest number of publications and the national contribution index. Among them, the United States has

the largest research scale and thus the highest contribution, with a national contribution index value of 20.37%; China ranked second globally, with an index value of 16.5%. The remaining countries contributed less than 10%. To some extent, this reflects the global competitive advantage of the United States and China in Metaverse research.

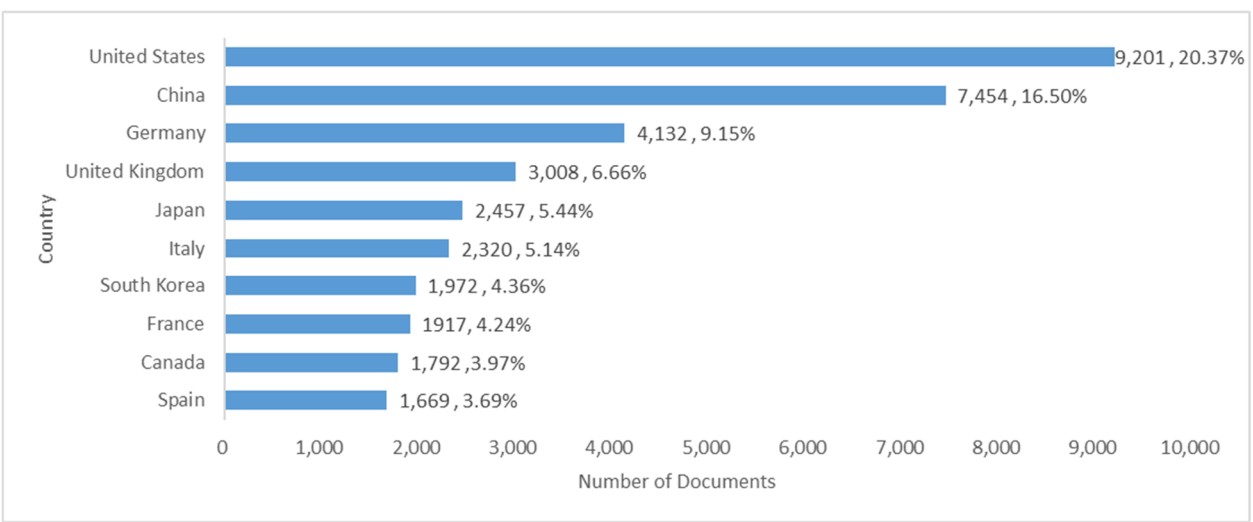

**Figure 5.** The national contribution index values of top 10 countries related to the Metaverse.

Figure 6 is presented for further observation of the annual publication changes of the 10 most highly contributing countries. In Figure 6, each of the color-coded branches corresponds to the top 10 high-contribution countries, with the width of the branches representing the country's current contribution degree. In the same time window, a wider corresponding width indicates a higher contribution degree of the country.

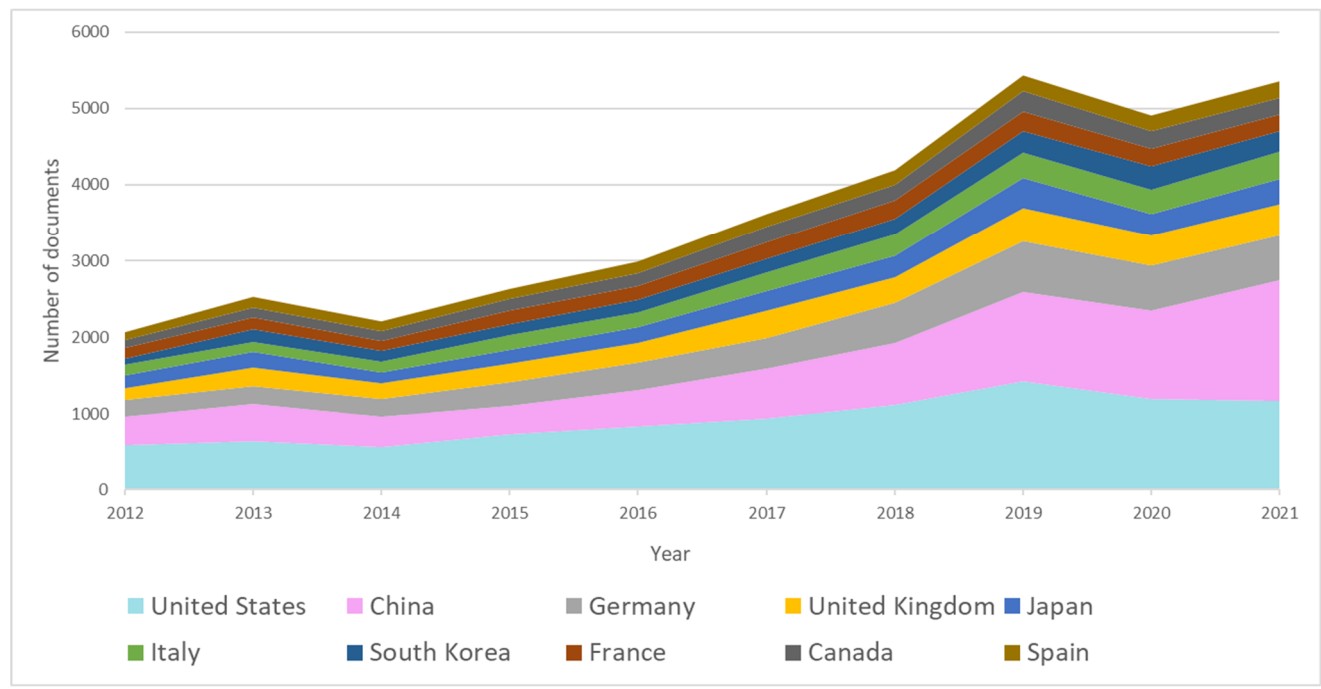

**Figure 6.** The changing contribution trends of the top 10 countries related to the Metaverse.

The total width of "rivers" was narrower in the early years and wider in the most recent years, indicating that the total number of annual publications of the top 10 contributing

countries is increasing in general. The total river width reached its maximum in 2019 and narrowed slightly in the following two years, which has been stated above as being due to the cancellation or postponement of some conferences or a lag in database inclusion. Among the branches, the sum of the width of the two branches representing the United States (light blue river) and China (pink river) is nearly half of the total river width in most time windows, indicating that the two countries are the main contributors to the Metaverse field. From the changes in the width of branch rivers, the width of branch rivers representing China increased significantly with time after 2016, indicating that the growth rate of China's publications in recent years was higher than that of other countries, and the number of publications in 2021 exceeded that of the United States for the first time.

3.3.2. Analysis of International Collaboration

To further explore the main partnerships among countries/regions, a co-authorship visualization map of countries/regions was drawn with VOSviewer. The minimum document threshold of a country/region was set at 60, and thus, 63 countries/regions out of 774 remained as visualization items. Figure 7 presents the structure of the comprehensive collaboration network among countries/regions from 2012 to 2021, with the size of the circles representing the number of documents. The United States, the largest circle in the map, also links most countries/regions (the number of links = 60), which presents the widest range of cooperation with other countries/regions.

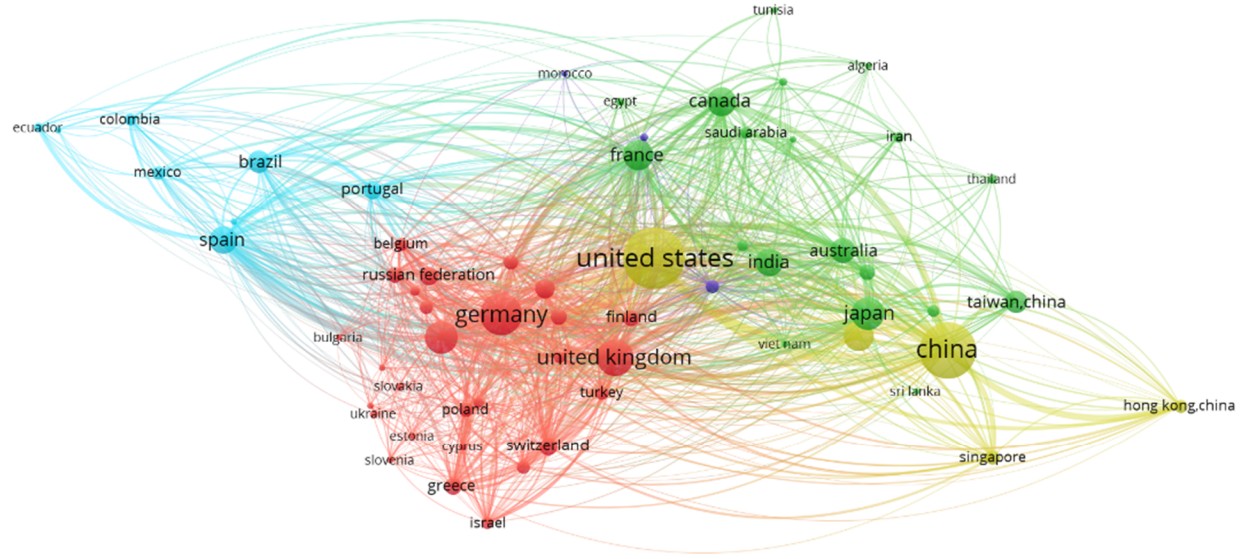

**Figure 7.** Co-authorship visualization map of countries/regions related to the Metaverse.

Five scientific camps in Metaverse research could be distinguished, as in Figure 7. Cluster 1 (red), led by Germany and the United Kingdom as the second and third most linked countries, includes 28 nodes in all. Most of the nodes in this cluster are European countries/regions, which indicates the tendency of cooperation with neighboring regions. The cluster containing the second largest number of countries/regions, colored in green, comprises 19 countries/regions, such as Canada, France and Japan, while the third one links 8 countries together, including 6 countries from South America and 2 from Europe. The fourth cluster, colored in yellow, obviously shows two "center" nodes as the United States and China, which are also the two largest sources of citations in statistics. The fifth cluster (purple) links three countries: New Zealand, South Africa and Morocco.

*3.4. National Differences: Comparison of the United States, China and Germany*

3.4.1. Comparison of Institutional Contributions

The explosion of the Metaverse has aroused widespread attention in various countries and regions. Due to the wide range of related technologies and application scenarios involved in the Metaverse, it is inevitable that research priorities, accumulated past research and technological advantages will vary among countries and regions. Therefore, a comparative analysis of countries is introduced in this section, and we further display and compare the publication status of institutions, co-occurrences and citation bursts of keywords in the three countries with the greatest contributions, which are the United States, China and Germany.

Table 4 shows the top 10 institutions in the United States, China and Germany with the highest number of publications in the past ten years, and it highlights the number of publications. It can be seen that the Chinese Academy of Sciences (CAS) and Technical University of Munich (TU Munich), the institutions with the highest number of publications in China and Germany, both rank much more highly than their domestic counterparts. Moreover, the gap between institutions is relatively small in the United States. In general, universities are the largest publishing institutions in the Metaverse field. The top 10 publishing institutions in the United States are all universities, while Chinese universities and German universities occupy 8 and 9 positions, respectively. For China and Germany, academic institutions are also represented well, with the Chinese Academy of Sciences and the German Research Center for Artificial Intelligence DFKI occupying the first and fourth positions, respectively, on the ranking lists of their respective countries. Specifically, one of the government departments in China, the Ministry of Education China, also has strong academic publishing performance, with 260 publications in the past ten years.

**Table 4.** Comparison of Institutional Contributions of the United States, China and Germany.

| Ranking | United States | China | Germany |
|---------|---------------|-------|---------|
| 1 | University of Southern California (288) | Chinese Academy of Sciences (458) | Technical University of Munich (327) |
| 2 | University of Central Florida (270) | Beihang University (327) | Rheinisch-Westfälische Technische Hochschule Aachen (191) |
| 3 | Stanford University (215) | Beijing University of Posts and Telecommunications (275) | Julius-Maximilians-Universität Würzburg (185) |
| 4 | Georgia Institute of Technology (184) | Shanghai Jiao Tong University (273) | German Research Center for Artificial Intelligence DFKI (166) |
| 5 | Carnegie Mellon University (181) | Tsinghua University (262) | Ludwig-Maximilians-Universität München (145) |
| 6 | Virginia Polytechnic Institute and State University (171) | Ministry of Education China (260) | Technische Universität Berlin (135) |
| 7 | University of Florida (165) | Beijing Institute of Technology (211) | Technische Universität Darmstadt (127) |
| 8 | Purdue University (157) | University of Chinese Academy of Sciences (198) | Universität Stuttgart (121) |
| 9 | Clemson University (149) | Zhejiang University (178) | Karlsruher Institut für Technologie (109) |
| 10 | University of Illinois Urbana-Champaign (125) | Huazhong University of Science and Technology (141) | Technische Universität Dresden (103) |

The University of Southern California, the top publishing institution in the United States over the past decade, ranks sixth globally, contributing to virtual humans, human computer interaction, user–computer interfaces and computer interfaces the most, in line with the research areas of the most published author, Gratch J, who published 32 articles and conference papers between 2012 and 2021, exceeding one tenth of the total amount. For instance, an NAO robot named Niki and a virtual human named Julie were controlled by a "Wizard of Oz" system to study factors impacting social influence [62]. After this, the Niki and Julie corpora were used to study influence and grounding in dialogue [63]. Social–emotional agents and negotiation are also research topics associated with virtual humans [64–66]. An Interactive Arbitration Guide Online (IAGO) platform was presented

as a tool for the design of human-aware agents used in negotiation [67]. The third most published author, Traum D, who cooperated with Gratch J, is also interested in virtual humans and human–machine interactions [68,69], while the second most published author, Bolas M, focuses on the interaction techniques adopted in virtual environments [70,71].

As the most prolific institution in the Metaverse field around the world, CAS's high-frequency research areas over the past decade include three-dimensional computer graphics, cloud computing, deep learning and augmented reality. The authors with the highest number of publications are Lv Z (15), Hou ZG (14) and Wang FY (11). Focusing on touchless motion interaction technology, Lv Z and his research group presented a hand-and-foot-based immersive multimodal interaction approach for handheld devices [72,73], and developed three primitive augmented reality games with eleven dynamic gestures [74], as well as a multimodal football game based on the multimodal approach as proof [73]. Hou ZG is committed to using the technology of virtual reality and the approach of BP neural networks in patients' recovery [75,76], and has presented a novel upper-limb rehabilitation robot and a 3D virtual reality simulator for core skills training in minimally invasive surgery [76–78]. Focusing also on the medical field, Wang FY stated that the ACP theory could be used in a parallel surgery system to significantly improve the efficiency and accuracy of surgical operations [79].

As the top publishing institution in Germany, TU Munich ranks fourth globally, from which the most published authors are Navab N(55), Klinker G(29) and Kellerer W(15). Navab N ranks eighth in the world in terms of publication volume, and augmented reality is a major area of concern in Navab's research, which is consistent with TU Munich. Navab N and his collaborators have studied the combination of AR technology and surgery, and they have reviewed the existing literature on augmented reality in robotic-assisted surgery [80]. They presented a method building on random forest to both provide the location of an instrument and the positions of the tool tips in real time [81], as well as providing an on-the-fly surgical support system that can be used in quasiunprepared operating rooms with the technology of augmented reality [82]. A head-mounted-display-based augmented reality system was also designed to guide an optimal surgical arm set-up via reflective AR [83].

### 3.4.2. Comparison of Keyword Co-Occurrence

To illustrate the research hotspots and perform a comparison among the three countries, keyword co-occurrence was further analyzed with VOSviewer in this study, and three visualization maps were drawn, as shown in Figure 8. In the process of mapping in Figure 8, the minimum number of occurrences of a keyword was set as 30 for all three countries, with 636, 474 and 255 keywords remaining for the United States, China and Germany, respectively. At the same time, the top 10 keyword occurrences were determined, as shown in Table 5, as a further supplement for the comparison among the three countries, highlighting the frequency of the occurrence of each keyword in parentheses.

As shown clearly in Figure 8, all three countries shared the same most frequently used keyword, "virtual reality", for which the total link strength was 45,601, 31,490 and 13,885, respectively. Among the visualization maps of the three countries, "virtual reality" is the central keyword, corresponding to all the red clusters, which means that it constitutes the largest cluster in any country, with 309, 139 and 124 items each. In concrete terms, the topics that the United States and Germany focus on are more consistent in the red cluster, including augmented reality as the main topic, which ranks second in terms of the total link strength in both clusters. By contrast, the topics of human–computer interaction and three-dimensional technology are more prevalent in the red cluster for China.

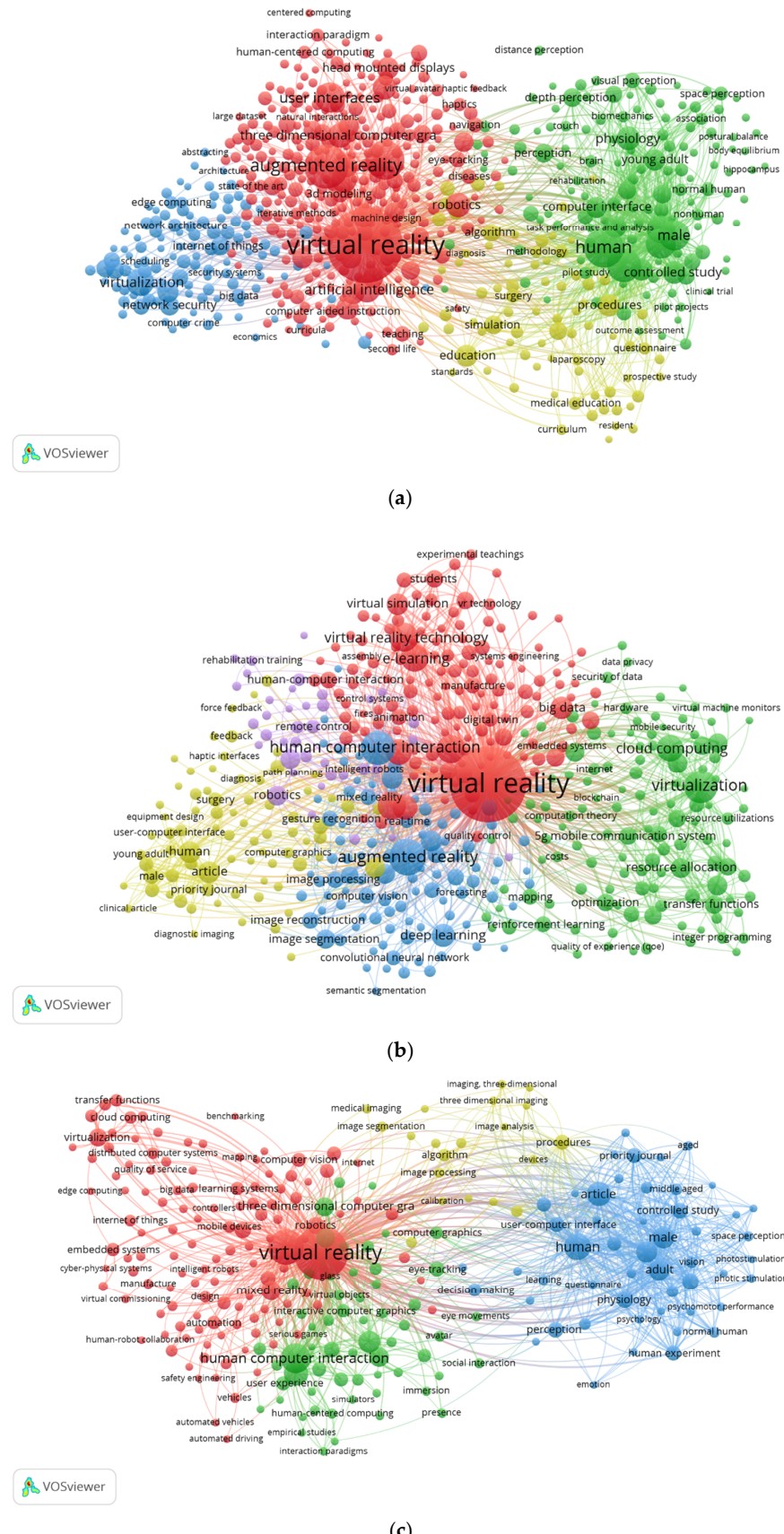

**Figure 8.** Comparison of keyword co-occurrence among 3 countries. (**a**) The U.S. (**b**) China. (**c**) Germany.

**Table 5.** Comparison of the top 10 keyword occurrences of the United States, China and Germany.

| United States | China | Germany |
|---|---|---|
| virtual reality (6465) | virtual reality (5687) | virtual reality (2792) |
| augmented reality (1898) | augmented reality (1071) | augmented reality (1048) |
| human (1454) | human computer interaction (870) | human computer interaction (709) |
| human computer interaction (1417) | virtualization (831) | human (545) |
| article (1059) | virtual reality technology (636) | user interfaces (452) |
| male (836) | artificial intelligence (569) | article (425) |
| female (801) | three dimensional computer graphics (567) | male (340) |
| user interfaces (765) | e-learning (536) | helmet mounted displays (336) |
| adult (700) | cloud computing (516) | female (330) |
| virtualization (635) | 3d modeling (453) | adult (313) |

On a country-by-country basis, four distinct colored clusters are clearly illustrated in the United States, with each representing a subfield. Following the red cluster, the green cluster gathers the second most keywords (134 items), which focus more on the user's experience, such as the user–computer interface. The cluster colored dark blue shows interest in "virtualization", with typical keywords including "network function virtualization", "virtual machine" and "virtualized environment". The fourth cluster, colored in blue, highlights the topic of "simulation", with keywords such as "computer simulation" and "simulation training". Another hotspot in this cluster is "imaging", following the keywords of "three dimensional imaging", "image enhancement", "image segmentation", "image processing" and "image reconstruction".

Different colors divide the distinct keywords in China into five clusters. The green cluster, including 129 nodes, focuses on the main domains of "virtualization" and "cloud computing", containing keywords such as "virtual addresses", "virtual machine monitors", "virtual network embedding", "cloud computing environments", "cloud computing platforms" and "cloud computing technologies". Another central cluster, in blue, includes 116 keywords, such as "user-computer interface", "helmet mounted displays", "human experiment" and "computer assisted surgery", referring to the use of technology in the Metaverse field to improve human well-being. Cluster 4 (yellow) comprises keywords such as "augmented reality", "deep learning", "learning systems" and "machine learning", while the purple cluster focuses on the study of robots and robotics, such as "robot programming", "intelligent robots" and "human robot interaction".

Appropriate labels for the four main clusters could be allocated to each of them by analyzing the main node circles in Germany. The green cluster contains the second highest number of nodes, with dominant studies on "human computer interaction", with keywords such as "helmet mounted displays", "human engineering", "user experience" and "behavioral research". Another sub-cluster (blue) focused more on "human experiments", and so that keywords such as "movement (physiology)", "brain computer interface" and "psychomotor performance" are included. Lastly, as the smallest cluster, imaging research such as "image segmentation", "image processing", "image enhancement", "medical imaging" and "three dimensional imaging" are more prevalent.

To summarize, although these three countries different from each other in terms of research scale and research hotspots in the Metaverse research, they still have some research interests in common. Led by research on "virtual reality", all three countries also place the focus on "augmented reality" and "human computer interaction", being among the five most frequent keyword occurrences listed in Table 5. According to the figure and the analysis above, the United States and Germany have both been interested in "imaging" studies in the past decade, while China focuses on "robot" studies, especially in the Metaverse area.

### 3.4.3. Comparison of Keyword Citation Bursts

Figure 9 shows the most rapidly growing topics in Metaverse research in the United States, China and Germany. Overall, the differences in the keyword citation bursts among the three countries are obvious, with almost half of the burst keywords being unique to each. Comparing them with each other, the keywords "big data", "cloud", "head mounted display", "immersion", "learning", "second life", "security" and "virtual agent" are exclusive to the United States, while the results for China uniquely highlight "image processing", "information technology", "intelligent system", "manufacture", "network security", "optimization", "robotics", "virtualization technology" and "wireless network". In Germany, the keywords "camera", "computer interface", "human robot interaction", "immersive virtual environment", "interactive computer graphics", "mixed reality", "mobile device", "sensory perception", "three dimensional", "user computer interface" and "user study" are specifically represented.

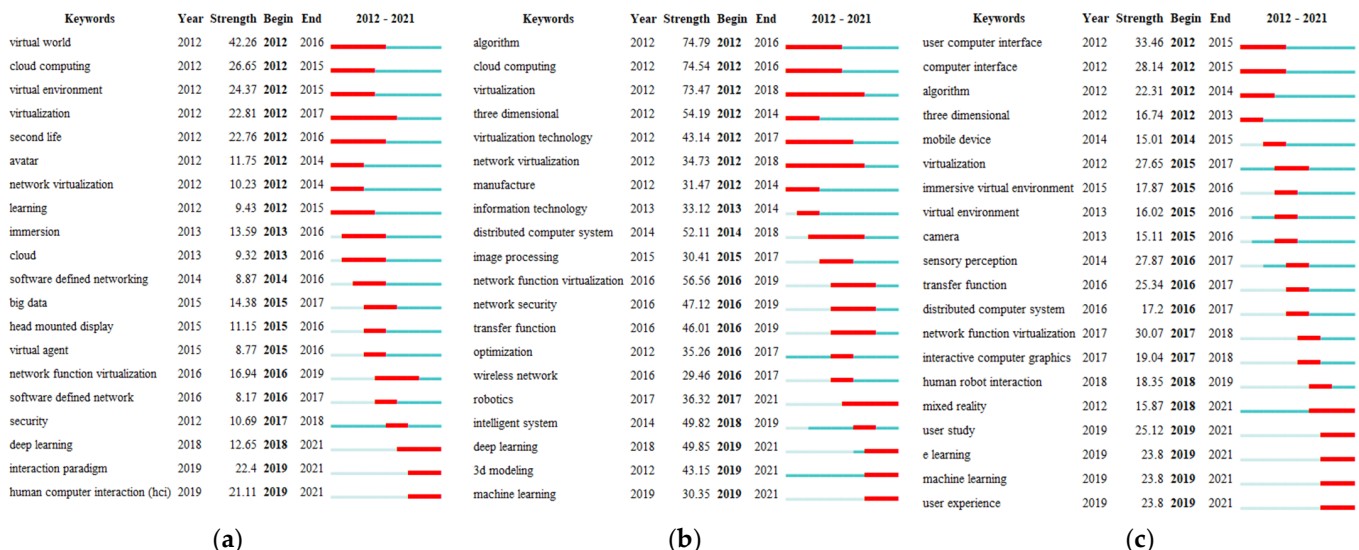

**Figure 9.** Comparison of the top 20 keywords with the strongest citation bursts among 3 countries. (**a**) The U.S. (**b**) China (**c**) Germany.

On the other hand, some of the same keywords emerge across countries. When compared with the figure of the top 20 keyword citation bursts globally, we can see that "virtualization" and "network function virtualization" appear in both the global figure and the three countries' figures, indicating that virtualization is a widespread research focus. "Virtualization" started to burst much later in Germany than in the other two countries, and it only burst in 2015–2017. "Network function virtualization" experienced a relatively consistent burst time, with the exception of Germany, during 2017 and 2018, while others showed abrupt changes between 2016 and 2019. For the rest of them, the United States and China share the burst keywords "cloud computing", "deep learning" and "network virtualization", in common with the worldwide result. China, Germany and the world all share the same hotspot of "distributed computer systems", although the emergence period is different. The global burst keywords "avatar", "human computer interaction", "software defined networking" and "virtual world" all appear in the results for the United States, showing the relatively high consistency of the two. Compared with the worldwide trend, the theme of three-dimensional technology in China is also relatively consistent, with it occurring in the earlier years with a short duration, while "3d modeling" burst in recent years. The keyword "algorithm" is also consistent with the worldwide trend both in China and in Germany, although the burst in Germany ended earlier than that worldwide.

When comparing the three countries, we can also find some similar burst keywords. For instance, both China and the United States showed a higher interest in cloud computing

in the early years. After 2016, both China and Germany paid great attention to the topic of transfer functions, while the hot topic lasted longer in China. Specifically, although the United States and Germany also presented the keyword virtual environment, there were obvious differences in the period of burst. The attention period was earlier in the United States, starting to burst in 2012 and ending in 2015, while it started in 2015 and lasted briefly until 2016 in Germany.

From the analysis of emerging hot topics by country, deep learning, interaction paradigms and human–computer interaction are new hotspots in the United States. Deep learning is also a new hotspot in China. Compared with the United States, it started to show a sudden increase one year later, in 2019. Robotics, 3D modeling and machine learning are three other hotspots in China that continue to the present. Compared with China, machine learning is also a new hot topic in Germany, while user study, e-learning and user experience are new themes emerging exclusively in Germany. Generally speaking, there are great differences in the emerging hot topics of concern among countries, which may be related to their inherent advantages in research and technology.

## 4. Conclusions

### 4.1. Conclusions and Limitations

The Metaverse is a fusion product that is currently expanding, and it integrates blockchain, digital twins, artificial intelligence, cloud computing, Internet of Things, high-speed networks and other technologies that have reached a certain height. It is an idealized virtual form, involving the intersection and convergence of multiple disciplines and technologies, which can be independently edited and reconstructed by users and ultimately coexist and interact with the real world. Given the early stages of its development, the Metaverse is increasingly attracting research. The aim of this study was to conduct a bibliometric analysis to examine the patterns and evolution related to the Metaverse by analyzing over 45,000 relevant papers published from 2012 to 2021 from the Scopus database, fully integrating the advantages of the bibliometric visualization software of VOSviewer and CiteSpace. Based on the above analysis, the researchers came to a number of conclusions, as follows:

First of all, as can be seen from the analysis of the evolution of the overall scientific publications, the Metaverse has become a field of extensive research during the past 10 years, corresponding to journal articles (36.15%) and conference papers (63.85%). The related publications have shown a distinct trend in recent years, while the greatest amount of scientific production was found in 2019 and 2021, with 6814 and 7062 publications, respectively. As can be seen from the rankings of publication contributors, the journal "Lecture Notes In Computer Science Including Subseries Lecture Notes In Artificial Intelligence And Lecture Notes In Bioinformatics", leading with 3060 documents, occupies the highest position among sources, while Billinghurst M is the author with the greatest output. When considering institutions, the highest-ranking institution is the Chinese Academy of Sciences (CAS), followed by Centre National de la Recherche Scientifique (CNRS) from France and Beihang University from China.

Secondly, in terms of countries/regions, the United States has contributed the most in the Metaverse field according to the number of scientific publications, followed by China and Germany. Specifically, the publication output on the Metaverse has increased exponentially since 2016 in China, with the number of publications being higher than that of any other country/region in 2021 for the first time, which is clearly reflected in the visualization of the changing contribution trends by country/region. Moreover, the three countries with the highest publication contributions, the United States, China and Germany, were chosen for comparison from the perspective of institutions. Among them, the scientific publication performance of institutions in both the United States and Germany is led by universities, namely the University of Southern California (288) and Technical University of Munich (327). The Chinese Academy of Sciences, as an academic institution, presents itself as the central producer, with 458 publications in China. Through country/region

cooperation analysis, VOSviewer separated the 63 analyzed countries/regions into five scientific camps that are, respectively, led by the United Kingdom, France, Spain, the United States and New Zealand. Differing from the overall research trend, the United States and the United Kingdom occupy the highest positions in terms of country/region collaboration.

Thirdly, based on the knowledge topic recognition and knowledge topic evolution path analysis of the research field, the research hotspots and frontiers were explored for the past 10 years. Co-keyword network mapping and keyword citation burst visualization were performed mainly using VOSviewer and CiteSpace. The network of the co-occurrence of author keywords is represented by three clusters and 401 nodes, where the term "virtual reality" has 30,794 occurrences relating to 400 terms. The analysis of keyword citation bursts implies that the themes of virtualization, cloud computing, virtual world, algorithm, network avatar and three-dimensional systems aroused many researchers' interest in the earlier years. It appears that distributed computer systems, network function virtualization, software-defined networking and edge computing are also established as research hotspots and frontiers in the Metaverse area. New research hotspots mainly focus on human–computer interaction, the user experience, 3D modeling, deep learning, human-centered computing, virtual reality technology, digital twin, e-learning and interaction paradigms, which is consistent with the trend of strengthening human–computer interaction and emphasizing user experience under Metaverse research.

In summary, the keyword co-occurrence analysis and keyword citation burst analysis pattern was repeated when performing the bibliographic coupling of three countries. Based on the co-keyword network analysis, it was found that the clustering characteristics of the Metaverse area are scattered in different research clusters among three countries; the co-keyword network visualization of the United States, China and Germany includes 4, 5 and 4 clusters, respectively, with the largest cluster of each being led by the central keyword "virtual reality". As for keyword bursts, some similar burst keywords emerge in both the global figure and the three countries' results, which are "virtualization" and "network function virtualization", while the differences in the keyword citation bursts among the three countries are still obvious, with almost half of the burst keywords being unique to each. To conclude, over time, the topic of Metaverse research has changed quickly among the three countries, with different branches synchronously thriving currently.

Some key limitations should be considered in this study. Concerning the methodological approach, the singular use of the Scopus database might have resulted in the omission of some important documents from other databases, e.g., Web of Science or Dimensions, and thus might have neglected certain contributions in the Metaverse field. Another reason for the lack of literature between 2012 and 2021 was the cancellation or postponement of some conferences and the lag in database inclusion, which would affect the overall development trend of 2020 and 2021. The data of the two years are continuing to increase currently, resulting in some literature in 2020 and 2021 that was not effectively included. This might cause new statistical changes in the clustering and abrupt trends of literature topics in the past 10 years. Additionally, due to the fact that our bibliometric work was carried out in 2022, and considering the comparability of the number of publications in each year, the Metaverse-related literature in 2022 was not included in the bibliographic database, which might result in missing some cutting-edge literature. The construction of the keyword database was mainly based on literature reading and expert consultation. Although keywords related to Metaverse technologies were included as much as possible, it was inevitable that some keywords were missing, which may have led to a lack of a few documents. On the other hand, it should be noted that the analysis, based on bibliometrics, had the characteristics of limited research objectives, specific research perspectives and quantitative research methods, which led to relatively effective research conclusions without other valuable information. For example, based on the limited literature classification in the Scopus database, the differences between theoretical research and empirical analysis could not be well distinguished. Furthermore, the comparative study between different databases could be conducted.

*4.2. Future Research Avenues*

Although it has aroused a lot of attention and research in recent years, the Metaverse is still full of infinite possibilities and endless speculation. Despite the existence of current literature, there is still a need for further research:

1. Economic systems have played pivotal roles in the Metaverse for various sectors, including Bitcoin, DeFi, cryptocurrencies, NFT, etc. [84]. Among them, NFT has the characteristics of particularity, uniqueness, pair-to-pair non-interchangeability, exclusivity, indivisibility and free tradability. By minting digital products into NFT, all stakeholders can control the ownership of digital products and share the economic value of the Metaverse. However, as a democratization content creation and commercialization tool, the NFTs give scammers and malicious content creators the opportunity to exploit the system by selling copies or low-quality NFTs [85]. With exaggerated price fluctuations, NFT also has great regulatory risks and is easy to be instigated by capital to seek profits in it, and new forms of monopoly may follow. At present, the economic ecology of the Metaverse is not mature and there are a lot of unsolved loopholes. Future research should focus on the risk of capital manipulation and industrial monopoly, and be aware of unreasonable industrial wind direction. The economic risks of the Metaverse world may be transferred from the virtual world to the real world, so it is more necessary to study how to build a safe and free Metaverse world in the future.

2. With the continuous development of the Metaverse industry and the scope of its application, several platforms have entered the market. Not only has it offered platforms for advertising and selling digital twins of real products, but it has also given large companies opportunity in the areas of recruiting and onboarding, such as Microsoft and its VR platform AltspaceVR [86]. Metaverse platforms are becoming popular in collaboration within virtual worlds, such as VoRtex designed for supporting collaborative learning activities [87], Nvidia's Omniverse Audio2Face2 and Meta's Oculus Lipsync3 for creating more realistic avatars [88]. In the gaming area, ZEPETO and Roblox are recognized as two of the most representative platforms of Metaverse, with future research needed to observe and interview platform users directly [89]. Future research is needed to answer the following questions: Of all these platforms, which is more likely to become the most publicly accepted platform in the Metaverse? How do we establish standards for the development and supervision of Metaverse platforms? How do we strengthen the link between the platforms and the overall coordination to achieve wider and more convenient connectivity of the virtual world?

3. Future Metaverse research should promote the ethical thinking of technology and the reform of social governance. In terms of practices and institutions of urban society, there is a lack of a theoretical basis and empirical evidence which is required to holistically assess the potential risks and hidden pitfalls of the transformative processes of smart urbanism [25]. It also opens security challenges in the massive virtual world where privacy attacks occur, such as eavesdropping by other platform users [85]. Moreover, ethical and moral issues such as integrity issues, publishing and disseminating false information, the problem of an unfavorable atmosphere and the infringement of intellectual property rights are urged to be solved [19]. The Metaverse demonstrates having a positive impact on social good, especially in terms of accessibility, diversity, equality, and humanity [3]. As part of scientific and technological development, it was expected that investments in the Metaverse should be justified by societal and ethical concerns more rather than global competitiveness and technological advancement. In future research, a mix of technical solutions and policies complying with the law need to be deployed, and different research objects need to be further related to provide a reference and basis for the development of the field of the Metaverse. A more holistic approach should be adopted to meet the future demands of Metaverse research in order to support the accelerated integra-

tion of related technologies and the further improvement of human production and living standards.

**Author Contributions:** Conceptualization, W.W., J.S. and X.Z.; methodology, J.S. and X.Z.; software, J.S. and Z.C.; writing—original draft preparation, J.S., W.W. and X.Z.; writing—review and editing, W.W., X.Z. and L.W.; visualization, J.S. and Z.C.; supervision, W.W. and L.W. All authors have read and agreed to the published version of the manuscript.

**Funding:** This research was based on the work supported by The General Program of the National Natural Science Foundation of China "The Studies on Evolution and Governance of Discipline Innovation Ecosystem Tracked".

**Institutional Review Board Statement:** The study was conducted in accordance with the Declaration of Helsinki and approved by the Institutional Review Board of Zhejiang University.

**Informed Consent Statement:** Not applicable.

**Data Availability Statement:** Not applicable.

**Acknowledgments:** We gratefully acknowledge the contributions of the Institute of China's Science, Technology and Education Policy (iCSTEP), Zhejiang University on this study.

**Conflicts of Interest:** The authors declare no conflict of interest.

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
