# Peer review of "Worldwide Overview and Country Differences in Metaverse Research: A Bibliometric Analysis"

_sustainability, doi:10.3390/su15043541_

Round 1
Reviewer 1 Report
Thank you for giving me the possibility to review this innovative and of interest work. It is about the study of the hotspots and frontiers of the academic research on the Metaverse based on bibliometric analysis from 2012 to 2021. The analysis is useful to provide to the reader an overview of the current status of the literature.
The Authors wrote a lot in the introduction of crypto currency... I'm of the opinion that a short introduction of the crypto currency, considering the topic, is useful, but I would like if this part is summarized.
I strongly suggest, instead, to implement the section that is about the potentiality of the metaverse in the healthcare system, for this purpose I suggest the following articles:
“Virtual reality aided therapy towards health 4.0: A two-decade bibliometric analysis.”
“The Metaverse: A New Challenge for the Healthcare System: A Scoping Review”
“Into the Spine Metaverse: Reflections on a future Metaspine”
Minor comments:
-Figure 4 show a trend from 2012 to 2022 while the analysis is until 2021. Please be consistent. If possible it should be interesting to include also 2022 considering how fast this field is evolving.
-At least a sentence about the future studies should be appreciated
Author Response
Dear review:
We are grateful for your effort reviewing our paper and your positive feedback. Here below we address our response for your suggestions:
Point 1: I strongly suggest, instead, to implement the section that is about the potentiality of the metaverse in the healthcare system, for this purpose I suggest the following articles: “Virtual reality aided therapy towards health 4.0: A two-decade bibliometric analysis.” “The Metaverse: A New Challenge for the Healthcare System: A Scoping Review” “Into the Spine Metaverse: Reflections on a future Metaspine”
Response 1: The introduction adds a supplement to the metaverse application and related bibliometric literature in the healthcare field, in which three articles suggested by the reviewer were fully referred to: Virtual reality aided therapy towards health 4.0: A two-decade bibliometric analysis; The Metaverse: A New Challenge for the Healthcare System: A Scoping Review; Into the Spine Metaverse: Reflections on a future Metaspine.
Point 2: If possible it should be interesting to include also 2022 considering how fast this field is evolving.
Response 2: For the timing of the literature selected, due to the fact that our bibliometric work was carried out in 2022, and considering the comparability of the number of publications in each year, the metaversal-related literature in 2022 has not been included in the bibliographic database. Relevant considerations and the possible absence of cutting-edge literature have been supplemented in the limitation section of the paper.
Point 3: The Authors wrote a lot in the introduction of crypto currency... I'm of the opinion that a short introduction of the crypto currency, considering the topic, is useful, but I would like if this part is summarized.
Response 3: As for the introduction of cryptocurrency, we put it into the introduction part to begin the narration from the emergence of the metaverse practice phenomenon. The time node of the narration of cryptocurrency, the launch of its related games and blockchain technology was all before the COVID-19 pandemic.
Point 4: At least a sentence about the future studies should be appreciated
Response 4: In terms of future studies, we added the avenues for future research in 4.2, including economic ecological construction and risk, platform supervision, ethical risk and social governance. In the part of economic ecological construction and risk, we focused on the summary and outlook of cryptocurrency and NFT.
Point 5: Figure 4 show a trend from 2012 to 2022 while the analysis is until 2021. Please be consistent.
Response 5: As for Figure 4, we are sorry for the mistake of year selection on CiteSpace during the earlier data analysis. The author re-analyzed the data and presented the graph. Since bibliometric analysis was completed in 2022, due to the completeness of the year and the comparability of the scale comparison of the number of publications in each year, literature in 2022 was not included in the literature database. However, the author cited some recent literature in 4.2 "Future Research Avenues" to look into the future.

Reviewer 2 Report
Give an explanation why the selected timespan is 2012-2021. Why start from 2012, why not from before? Give strong reasons accompanied by valid academic references
Give reasons why only Scopus is used as the database where the search is performed. Why isn't the Web of Science included? in some countries WoS is more recognized than Scopus.
Explain the meaning of the numbers in brackets in Table 2. Or at least provide an explanation in the table so that the reader can understand.
In the Keyword Analysis section, why don't you also include "virtual worlds", "mirror worlds", and "lifelogging"? As we know, those three keywords are included in the types of metaverse that have been widely referred to by researchers, even since the Metaverse Roadmap document was published.
You must also explain the reasons for choosing each keyword in Table 3, along with valid academic references.
Since you mentioned several Metaverse Platforms in the Introduction, for example, Meta and platforms from NVIDIA, then you should also analyze from the platform side used. To see which Platform is most widely accepted and will likely become the standard Platform for Metaverse development. As we know, currently there are too many platforms, so there are too many "Verses", and in the end, it becomes the Multiverse. The hope for the future is that the Metaverse is just like the internet, we just need to be connected to the Internet and make direct interactions. Thus the contribution of your manuscript will be more meaningful
Author Response
Dear review:
We are grateful for your effort reviewing our paper and your positive feedback. Here below we address our response for your suggestions:
Point 1: Explain the meaning of the numbers in brackets in Table 2. Or at least provide an explanation in the table so that the reader can understand.
Response 1: The meaning of the numbers in brackets in Table 2 has been supplemented using changing form.
Point 2: Give reasons why only Scopus is used as the database where the search is performed. Why isn't the Web of Science included? in some countries WoS is more recognized than Scopus.
Response 2: As for the selection of databases, the existing literature has compared WoS database with Scopus database, and reached a relatively consistent conclusion, namely, the number of papers obtained through the two databases, the number of citations and the ranking of the papers in each country were highly correlated. At the same time, the Scopus database covers a larger amount of literature. Therefore, Scopus database was selected in this study. The above considerations were responded to in "2.1. Data Collection" and the relevant literature was supplemented.
Point 3: In the Keyword Analysis section, why don't you also include "virtual worlds", "mirror worlds", and "lifelogging"? As we know, those three keywords are included in the types of metaverse that have been widely referred to by researchers, even since the Metaverse Roadmap document was published. You must also explain the reasons for choosing each keyword in Table 3, along with valid academic references.
Response 3: As for the selection of keywords, we added a list of documents referenced, including the main reports(like White Paper on China's Metaverse), review literatures (like A Survey on Metaverse: the State-of-the-art, Technologies, Applications, and Challenges; All One Needs to Know about Metaverse: A Complete Survey on Technological Singularity, Virtual Ecosystem, and Research Agenda; 3D Virtual worlds and the metaverse: Current status and future possibilities; Virtual world, defined from a technological perspective and applied to video games, mixed reality, and the Metaverse), and other key literatures (like What is a metaverse; Metaverse; Fusing Blockchain and AI with Metaverse: A Survey) that during the construction of the keyword library。Among them, the keyword "virtual world" has previously been included in the "Metaverse perspectives" section of the keyword library in Table 3. Since the construction of the keyword database is mainly based on literature reading and expert consultation, it is inevitable that there will be some omissions and cannot be covered completely. In this regard, the author wrote the limited part. Thanks to the expert opinions sincerely.
Point 4: Give an explanation why the selected timespan is 2012-2021. Why start from 2012, why not from before? Give strong reasons accompanied by valid academic references.
Response 4: For the timing of the literature selected, the reason for choosing to search since 2012 is mainly for two reasons. One was that the first non-functionable token (NFTs) Colored Coins in history came out, serving as an economic bridge between Metaverse and the real world, leading the increasingly integration of the real world and the digital world. Another reason was, the emergence of NFTs enables the Metaverse to expand various social meanings (such as fashion, activities, games, education, office, etc.) on the basis of immersive interaction, which is different from the early concept of the metaverse.
Point 5: Since you mentioned several Metaverse Platforms in the Introduction, for example, Meta and platforms from NVIDIA, then you should also analyze from the platform side used. To see which Platform is most widely accepted and will likely become the standard Platform for Metaverse development. As we know, currently there are too many platforms, so there are too many "Verses", and in the end, it becomes the Multiverse. The hope for the future is that the Metaverse is just like the internet, we just need to be connected to the Internet and make direct interactions. Thus the contribution of your manuscript will be more meaningful.
Response 5: On metaverse platforms, "4.2Future Research Avenues" gave examples of current avenues in business, games, and education. We also made discussions for future research, including Of all these platforms, which is more likely to become the most publicly accepted plat-form in the metaverse? How to establish standards for the development and supervision of metaverse platforms? How to strengthen the link between the platforms and the overall coordination, to achieve a wider and more convenient connectivity of the virtual world?

Reviewer 3 Report
1. The article presents a study on worldwide overview and differences that exists followed by analysis.
2. The paper is balance towards to theoretical aspect highlighting in the existing information in summarized forms with tables and figures.
3. Representation of Table 1 could be though of in a different way of presentation.
4. The paper doesnt have any architectural illustrations, as the nature of the paper is trending towards representation of known information and summarized sequentially.
5. The conclusion of the paper should be more precise in a summarized form and need to be restructured in shorter form. In current form, it is more lenghty.
6. Good number of references were seen in the paper and cited properly as well which is good.
Author Response
Dear review:
We are grateful for your effort reviewing our paper and your positive feedback. Here below we address our response for your suggestions:
Point: The conclusion of the paper should be more precise in a summarized form and need to be restructured in shorter form. In current form, it is more lenghty.
Response: The first draft of the paper does have the problem that the conclusion part is more lenghty. Thanks for your precious comments. For this, we divided the conclusion into "4.1. Conclusions and Limitations" and "4.2. Future Research Avenues". At the same time, we added discussion on "4.2. Future Research Avenues," including economic ecological construction and risk, platform supervision, ethical risk and social governance.
